# Monopolar and Bipolar Combination Injuries of the Clavicle: Retrospective Incidence Analysis and Proposal of a New Classification System

**DOI:** 10.3390/jcm10245764

**Published:** 2021-12-09

**Authors:** Mustafa Sinan Bakir, Roman Carbon, Axel Ekkernkamp, Stefan Schulz-Drost

**Affiliations:** 1Department of Trauma and Reconstructive Surgery and Rehabilitative Medicine, Medical University Greifswald, Ferdinand-Sauerbruch-Straße, 17471 Greifswald, Germany; axel.ekkernkamp@uni-greifswald.de; 2Department of Trauma Surgery and Orthopedics, BG Hospital Unfallkrankenhaus Berlin gGmbH, Warener Straße 7, 12683 Berlin, Germany; stefan.schulz-drost@helios-gesundheit.de; 3Department of Pediatric Surgery, University Hospital Erlangen, Krankenhausstraße 12, 91054 Erlangen, Germany; roman@carbon-von-frankenberg.de; 4Department of Trauma and Orthopedic Surgery, University Hospital Erlangen, Krankenhausstraße 12, 91054 Erlangen, Germany; 5Department of Trauma Surgery, Helios Hospital Schwerin, Wismarsche Str. 393-397, 19049 Schwerin, Germany

**Keywords:** clavicular combination injury, acromioclavicular joint dislocation, sternoclavicular joint dislocation, clavicle fracture, floating clavicle, big data analysis, routine data, classification system

## Abstract

Clavicle injuries are common, but only few case reports describe combined clavicular injuries (CCI). CCI include combinations between clavicular fractures and acromioclavicular/sternoclavicular joint dislocations (SCJD). We present the first general therapeutic recommendations for CCI based on a new classification and their distribution. A retrospective, epidemiological, big data analysis was based on ICD-10 diagnoses from 2012 to 2014 provided by the German Federal Statistical Office. CCI represent 0.7% of all clavicle-related injuries *(n* = 814 out of 114,003). SCJD show by far the highest proportion of combination injuries (13.2% of all SCJD were part of CCI) while the proportion of CCI in relation to the other injury entities was significantly less (*p* < 0.023). CCIs were classified depending on (1) the polarity (monopolar type I, 92.2% versus bipolar type II, 7.8%). Monopolar type I was further differentiated depending on (2) the positional relationship between the combined injuries: Ia two injuries directly at the respective pole versus Ib with an injury at one end plus an additional midshaft clavicle fracture. Type II was further differentiated depending on (3) the injured structures: IIa ligamento-osseous, type IIb purely ligamentous (rarest with 0.6%). According to our classification, the CCI severity increases from type Ia to IIb. CCI are more important than previously believed and seen as an indication for surgery. The exclusion of further, contra-polar injuries in the event of a clavicle injury is clinically relevant and should be focused.

## 1. Introduction

Bony and ligamentous injuries to the clavicle are relatively common, but the number of studies depends on the frequency of the respective injury entity: in common clavicle midshaft fractures and lateral injuries, more literature is detectable than in the rare medial ones [1]. A special and rare entity is the clavicular combination injury (CCI) [2,3,4,5]. Regarding CCI, only marginal data describing this type of injury have been published so far, and only a few case reports and a small case series have been published [2,6,7]. The maximum degree of a combination injury in the sense of the highest degree of instability occurs in a bipolar clavicular dislocation, which is also referred to as the “floating clavicle” [8,9,10]. Biomechanically, all CCIs generally cause an unstable situation for the clavicle, which is analogous to the bipolar clavicular dislocation known as the “floating clavicle”. It is a widespread assumption that a ligament disruption and fracture do not occur in the same case, since, either a fracture occurs, or the force disrupts the joint and ruptures ligaments [3,9,11]. While the CCI is usually detected at least at one end or occurs as a single injury, due to the rarity of the combined injury, a coincident injury at the clavicle is often not expected [3,9,11,12,13]. Therefore, the risk of missed injuries with this entity has proven to be particularly high, and a high number of unreported cases can be assumed [12,14,15,16,17,18,19]. A combination injury diagnosed secondarily, however, can have serious consequences in the short-term with prolonged pain and restricted mobility, and in the long-term with post-traumatic joint arthritis, which is highly likely in the event of complete instability [15,16,17,20].

The sparse literature that addresses this topic presents a minimal amount of data on incidence and age distribution and mostly addresses the selected types of combination injuries to the shoulder girdle but is not comprehensive [14]. We assume that CCIs represent an underrated entity on the shoulder girdle. Therefore, our study primarily aimed to substantiate the hypothesis that CCIs are more common than expected although they have not been well-described in the literature so far. Based on this goal, this study thoroughly reviews combined injuries between the clavicle and the corresponding joints for the first time in order to demonstrate the distribution of the different combinations and to derive a classification for the CCI. Therefore, this retrospective, epidemiological, big data analysis will suggest the first therapeutic recommendation for CCI and will propose it on the basis of our new classification.

## 2. Materials and Methods

The data from the International Classification of Diseases (ICD-10) coding of all German hospitals for a period of three years (2012–2014) were analyzed retrospectively after they were made available by the German Federal Statistical Office/Destatis [21,22]. All clavicular injuries of the shoulder girdle were examined as an anatomical functional unit. Therefore, injured persons with the combinations based on ICD-10 codes S42.01, S42.02, S42.03 (medial, midshaft, and lateral clavicle fracture [MCF, MICF, and LCF], respectively) in addition to S43.1 (acromioclavicular joint dislocation [ACJD]) and S43.2 (sternoclavicular joint dislocation [SCJD]) were considered. The subgroup of medial clavicular injuries (MCI) includes MCF and SCJD, while LCF and ACJD were considered as lateral clavicular injuries (LCI). Injuries were explicitly contra-polar if they occurred at one end (e.g., medial) and at the opposite pole (e.g., lateral).

Data from Destatis were included in the form of main and secondary diagnoses. Patients of all ages were evaluated. Combinations of purely bony injuries were excluded (combinations of MCF, MICF, and LCF, respectively). Chronic instabilities were another exclusion criterion, as we wanted to relate our analysis exclusively to acute, traumatic injuries. The CCI were classified according to the type of combination (monopolar versus bipolar) and examined for incidence, distribution of the individual types, and age. Based on the proposed new classification, we compiled therapy options and included some of the strategies described from individual case reports, evaluated them, and created an overview of treatment recommendations with reference to our classification.

The study was registered with the German Clinical Trials Register (DRKS; DRKS00017018) and was performed in accordance with the ethical standards laid down in the 1964 Declaration of Helsinki after approval by the local ethics committee (University Medicine Greifswald, BB 007/19). Informed consent was obtained from all individual participants included in the study, including participation and publication of their imaging results and of potential identifying information. However, the majority of the information was purely retrospective and based on anonymized data provided by the German Federal Statistical Office.

Statistical analysis was performed using the SPSS software (IBM Corp., SPSS Statistics for Windows, Version 26.0. Armonk, NY, USA). The associations between types of injury entities and further analyses were tested using Pearson’s chi-square test, and in case of low cell frequencies (for example, single cell number < 5) via Fisher’s exact test with an alpha-level of 0.05. No alpha adjustment for multiple testing was conducted due to the explorative character of the analysis.

## 3. Results

### 3.1. Incidence and Subtypes of Combined Injuries

For comparison of routine Destatis data, a total of 114,003 cases of the five clavicle-associated injuries were analyzed with special focus on the combinations between the five individual diagnoses (Figure 1, Table 1). We understood combinations that consisted exclusively of fractures without the involvement of a clavicle joint injury to be synonymous with multi-fragmentary clavicle fractures. Therefore, these were excluded from the group of CCI. The total number of the two types of combination injuries was 814, so 0.7% of all cases were combined injuries.

Our collective showed an age distribution with incidence peaks in the life stages from 40 to 45 years and a somewhat smaller peak from 20 to 35 years (Figure 2a,c). From the age of 55 years and older, however, the number of cases decreased rapidly and continuously. A similar tendency can be seen with the MICF in which the peak in younger adulthood/adolescent age is earlier and more pronounced (Figure 2b,c).

The most common combinations (Table 1, Figure 3) were those between LCF and ACJD (*n* = 570, 70.0% of all CCI) and MICF and ACJD (*n* = 106, 13.0% of all CCI).

The proportion of combinations in medial clavicular injuries (MCI) differed significantly (*p* = 0.004) by 1.0% (138/14,264) from the proportion with lateral clavicular injuries (LCI) of 1.3% (739/58,583). In particular, the proportion of explicitly contra-polar injuries showed a significant difference (*p* < 0.001) between medial and lateral injuries (Table 1 and Table 2): the proportion of combined, additional, lateral injuries in MCI was 0.4% (63/14,264) and thus four times as high as that of combined, additional, contra-polar medial injuries in LCI with 0.1% (63/58.583). In the latter case, however, this occurred in the opposite way, namely the predominance of bipolar combination injuries in MCI. The proportions of CCI in relation to the respective injury entity differed significantly from one another with SCJD showing by far the highest proportion of combination injuries with more than every 10th case of SCJ injury (Table 2).

### 3.2. Classification of Clavicular Combination Injuries

Based on these data, we classified CCIs in more detail involving three parameters. We divided these injuries into two different types depending on (1) the polarity of the injuries (monopolar/bipolar). The more common type I was considered in the sense of a monopolar CCI and the less common type II in the sense of a bipolar CCI (Figure 4 and Figure 5). Type I was further differentiated depending on (2) the positional relationship of the injuries into type Ia, with monopolar injuries directly at the respective pole end, and into type Ib, with monopolar injuries including a MICF. Type II was further differentiated, depending on (3) the injured structures, resulting in type IIa with ligamento-osseous injuries and type IIb with a combination of purely ligamentous injuries. Type I injuries are by definition all ligamento-osseous injuries due to the associated types. We define the polarity of type I injuries as “L” for lateral and “M” for medial.

A “tripolar” CCI is also conceivable as a worst-case scenario, which would represent type III in our classification. In the case of these injuries, an MICF would also be present in addition to the bipolar injuries of type II (most likely of type IIb). This process would affect all three columns in the sense of a tripolar comminuted dislocation fracture. However, this type has not been shown in our analysis, and it has not been described in the literature to date as far as we know.

With regard to the clavicular joint dislocations, the classification is restricted to only acute, traumatic injuries for our classification. We exclude chronic instabilities from the consideration of the classification. In addition, we refer exclusively to complete dislocations resulting in an unstable joint. Therefore, we only included higher-grade CJI with complete rupture of the corresponding ligaments, so it is synonymous with SCJD type Allman III and ACJD at least type Rockwood IIIb [20,23]. In the case of clavicle fractures, all types were included (regardless of the type, according to the classification, such as the LCF according to Jäger and Breitner or the MCF according to Bakir [24,25]).

Type I is almost 12 times as common as type II, which is only present in less than one in ten cases (Table 3). The monopolar lateral injuries account for a much larger proportion of the total CCI than the medial ones. The purely ligamentous, bipolar injuries of type IIb constituted the smallest proportion.

A comparison of the age distribution between types I and II shows an almost identical age pattern except for the missing peak of type II injuries between 25 and 35 years (Figure 6).

### 3.3. Treatment Algorithm of Clavicular Combination Injuries

For our group, the CCIs that are presented are basically considered as an indication for surgery (Table 4). Conservative therapy should only be reserved for cases with serious contraindications. Since the classification only includes high-grade clavicular joint dislocations, these injuries presuppose a high degree of instability. According to our new classification, the severity of the combination injury increases with increasing type from type Ia to IIb. This process can be seen as both a decrease in the incidence and in the selection of the osteosynthesis method, which is still more urgently indicated and more complex (Table 3 and Table 4, Figure 7). The lower, monopolar type Ia, for example, can also be treated with a single surgical method, whereas the higher-grade type II CCI always require a combination of more than one surgical procedure due to the bipolarity.

## 4. Discussion

### 4.1. Incidence and Subtypes of Combined Injuries

Our findings demonstrate that the combination injuries of the shoulder girdle, which have so far hardly been reported, seem to occur more frequently than previously assumed. So far, this entity has only been mentioned in case reports, which suggests that these injuries are rare since no definitive incidence of these injuries has been reported in the literature yet [2,3,4,10]. However, with 0.7% of clavicle injuries, these injuries make up a small, but not to be underestimated proportion, since a clavicle injury is relatively common overall [1,26,27,28]. This proportion is similar to a group that is as comparable as possible in a series of 614 clavicle fractures from which 0.8% had segmental injuries as the closest approximation to bipolar combination injury [29]. Due to the large total number of clavicular injuries, we believe that the knowledge gap concerning CCIs should urgently be closed.

Particularly in lateral shoulder girdle injuries, combinations are significantly more common than in medial ones. This finding is due in particular to the higher absolute total number of lateral injuries [1,30,31]. The lateral type Ia injuries in terms of combinations of LCF and ACJD are also generally considered as the most common subtype of CCI, which was confirmed in our analysis [17,32,33]. In contrast, SCJD are the most frequently associated single injury with a further injury to the shoulder girdle in more than 10% of all cases. This finding coincides with preliminary examinations that suggest and have proven a high proportion of clavicular in addition to further additional injuries in general in MCI [24,34,35]. In a current registry study, more than every 10th case of a clavicle joint injury (SCJ and ACJ) in severely injured patients had an additional CF, a finding that is quite similar to our data [36]. This collective showed that 0.1% of all patients altogether were affected by a CCI [36]. Therefore, especially in everyday clinical practice, it is essential to further exclude contra-polar injuries, especially in the case of an injury in the sternoclavicular region but also in the case of any clavicular injury [15,37]. Thus, underdiagnosis or delay can negatively influence the effectiveness of treatment and economy in a negative way [38].

As already published in case reports, when diagnosing a suspected mono-injury of the shoulder girdle, there is a risk of overlooking (or misinterpreting/misdiagnosing) further trauma consequences in this area [5,9,12,13,16,18]. It is often assumed that the trauma impact only develops its full force at one point and causes one single injury since the exact mechanism that is responsible for a combination of clavicle injuries is still unknown [3,39]. We therefore assume that a relevant number of CCIs are overlooked in everyday clinical practice or are only diagnosed secondarily after a delay, which would also provide a possible explanation for the limited number of studies in this regard [9,17]. Even in a case when the ACJD with a lateral CCI type Ib did not always seem to be symptomatic (in the short-term follow-up), we believe as do other authors that a missed CCI can, however, seriously cause unnecessary pain and post-traumatic osteoarthritis [14,15,16,17,18]. To avoid these potential consequences, a conceivable alternative would be the recommendation of a CT scan or an MRI in case of a certain grade of trauma mechanism in order to be able to reliably prove or exclude the presence of further injuries in the sense of a CCI. A recommendation in this regard, as recently made for the sternoclavicular region, cannot yet be made and should be analyzed in further studies [38]. The way a CCI influences the outcome compared to mono injuries is not known, and as with determination of the prognoses of the individual CCI types, should be the subject of further investigation [14].

Age distribution is largely similar to the distribution of MCI and clavicle injuries in general [1,24,35,40]. Although some authors have described that a greater tendency of clavicle fractures occur in younger adulthood, the age distribution of more complex clavicular injuries, such as CCI and as MCI, appears to be similar [1,27,41]. The age distribution seems to be most likely due to a higher risk-taking behavior in the mainly affected age groups, which would be in line with previous investigations, but still has to be proven [1,42].

A sub-classification of CCI has not yet been designated [3]. Our new classification of CCI should provide a tool for their consideration and treatment as rare injuries due to a lack of standard concepts that often lead to uncertainties in the choice of therapy. In order to present a consistent concept, we excluded multi-fragment or segmental CFs, which, according to our definition, do not represent a CCI in the classic sense. Since the coding using S42.09 according to the ICD-10 is synonymous with multiple CF but was used inconsistently, these injuries were also excluded in connection with the combinations between MCF, MICF, and LCF [22]. Nonetheless, multi-fragmentary/segmental clavicle fractures require special attention but have already been considered in the past in contrast to consideration of CCIs [43,44,45].

### 4.2. Classification of Clavicular Combination Injuries

In our opinion, the various subtypes of the CCI classification belong to a common group of one entity. These combinations of clavicle-associated injuries are treated with similar surgical procedures or the same approaches. Via the shoulder girdle, the clavicle represents the most important connection between the upper extremity and thorax [34,46]. Classification systems for fractures or dislocations are important for accurately identifying injury patterns, categorizing management problems, and guiding treatment decisions [47]. In the current classifications, the CCI are usually not accurately classified or not adequately represented due to a lack of accuracy with which the previous classification systems describes the true pathologic process [5,17,47]. A uniform designation is currently not used for bipolar clavicle injuries so that these injuries were named with different synonyms in the past, such as complete/bipolar/pan-clavicular dislocation, and floating clavicle among others [10].

The various degree of severity of the individual injuries was deliberately not included in the classification. The integration of the classifications according to Rockwood or Allman with consideration of the degree of displacement and description of the injured structures would have meant a higher precision in the specification of the injuries [20,23]. However, the authors believe that this would have more significantly complicated the classification in terms of its manageability. Since the classification only includes higher-grade, complete, and unstable joint dislocations of the SCJD type Allman III and ACJD at least from type Rockwood III(b), these are usually regarded as an indication for surgery even in case of a mono injury [23,34,48,49,50,51,52]. However, the statement on the indication for surgery should be restricted because a clearly defined standard therapy for SCJD is missing. Some authors advocate a conservative approach in some cases even for Allman III injuries and the indication for surgery for ACJD, especially for Rockwood III injuries, is in some cases still controversial [34,48,50,52,53,54,55]. However, the opinion that a Rockwood III ACJD is an indication for surgery is common in Germany and is in line with the results of an online survey in which this type is mainly treated surgically [52].

All types of clavicle fractures, regardless of the type according to the classification of the LCF according to Jäger and Breitner, or Neer, or Arbeitsgemeinschaft für Osteosynthesefragean/Orthopedic Trauma Association (AO/OTA), or the medial fracture according to Bakir or Robinson were included in this new classification [24,25,31,33,56]. Therefore, there is the analogous possibility that un-dislocated fracture types also are included in the consideration, but which would have been treated conservatively as isolated single injuries. Nonetheless, the authors claim that the CCI described are associated with further and additional instability due to the ipsilateral combination of injuries and should therefore be surgically restored to stability [5,51].

We postulate that, according to our classification, as the type of injury entity increases, so does the severity/complexity of the combination injury. Type Ia injuries often could be treated with a single osteosynthesis such as a hook plate when combining LCF/ACJD. In contrast, in the case of type II injuries, maximum instability in the sense of a floating-like clavicle or even a “real” floating clavicle (type IIb) can be found, which requires a combination of osteosynthesis procedures as shown in Figure 7 [51].

A type III injury or “tripolar” comminuted dislocation fracture in which all three columns were injured medially, in the midshaft and laterally, is not shown in the data or the literature, but would theoretically represent the worst case scenario in our classification.

### 4.3. Treatment Algorithm of Clavicular Combination Injuries

CCIs are rare, which we were able to show in our analysis. Therefore, no studies evaluating the therapy recommendations presented and proposed by us have been done. With regard to combined injuries involving SCJD, a large number of different therapy options that have been published in individual case reports can be found [51]. However, these examples are related almost exclusively to isolated injuries of the SCJ since even the care of this entity as an isolated single injury does not yet have standardized treatment algorithm and is very heterogeneous [24,34,48].

A high variety of treatment approaches with regard to CCI, including nonsurgical, surgical, and hybrid management of the two respective parts of this injury entity can generally be found [2,45]. The subgroup of surgical procedures was also heterogeneous without a predominant consensus [2,10]. The treatment strategy is mainly related to the involvement of the respective structures and the degree of severity [51]. In most cases of MCI, this procedure was carried out in line with published recommendations [24].

The illustrated arthrodesis of the SCJ is only intended as a temporary arthrodesis. This process then logically goes hand in hand with further surgery as part of an implant removal and then can also be associated with the surgical risks of a second operation. A necessary implant removal is used in many therapy options for clavicle joints, but exceptions are available that do not require implant removal, for example, an arthroscopic technique surgery using TightRope (Arthrex Inc., Naples, FL, USA) or Dog Bone Button technology (Arthrex Inc., Naples, FL, USA), or polydioxanone suture (PDS) banding. In addition, as with any arthrodesis, the SCJ arthrodesis can be associated with short-term and subsequent complaints so that the appropriate selection of therapy method should be carefully considered. The same steps apply when considering individual patient factors, such as accompanying illnesses, patient age, or the athletic and/or physical demands of the patient, for example, overhead work [55]. The correct treatment selection in the correct situation is important; not only does the injury morphology naturally play an important role in the selection of the surgical procedure but the importance of the surgeon’s familiarity with the procedure should not be underestimated since the outcome also depends on the training of the physician with respect to familiarity with the procedure [55]. We know, for example, that a tendency toward better results and higher patient acceptance is seen with arthroscopic procedures for ACJD as is described in the current literature, but no significant clinical differences in outcome have been demonstrated so far [50,55]. Nevertheless, this procedure is not yet widely performed and is less common, especially with non-specialists in shoulder surgery [50,52].

Type II(b) injuries are the least observed CCI but are accompanied by a clear therapy recommendation due to the presence of a maximum version of the floating(-like) clavicle [9,51]. Whether a bipolar surgical repair via TightRope should be performed or whether an alternative therapy method should be chosen for one of the clavicle poles (such as a combination of TightRope at SCJ and hook plate at ACJ) remains to be discussed [2,12,51,57,58,59]. At minimum, a bipolar rigid restoration can have consequences in the event of renewed trauma, even if it is only slight, since this mechanical conduction causes stress due to the lack of elasticity and creates a predetermined breaking point at the clavicle [9]. Further research and/or biomechanical analyses should show whether a rigid fixation option on one of the poles is a requirement for stability in order to enable a safe contra-polar TightRope fixation.

The fact that surgery is more often necessary in the case of combined injuries could be due to the fact that the necessary stability has to be regained [5]. Since massive instability prevails in the sense of a floating(-like) clavicle, no adequate healing would otherwise be possible so that anatomical reduction and fixation is essential [5,37,51]. In the literature, no stringent therapy standard has yet been established for combined injuries to the shoulder girdle so that a comparison is only possible with a large number of surgical methods obtained from case reports, which were not shown to be homogeneous [5,45,60]. We attribute the surgical indication itself to the multidirectional unstable situation that frequently occurs in CCIs.

The fact that no surgical treatment of the involved structures was carried out in the event of a CCI is rather the exception and was only shown in a few case reports [10]. In one case report, however, the second part of the injury was only found on post-operative radiographs and probably did not influence the outcome, but this outcome was only re-examined at a short-term follow-up, and osteoarthritis that may have developed later could not be ruled out [18]. In other cases, the second diagnosis was also delayed or abnormalities in the healing process, such as minimal distal clavicle osteolysis or ACJ (re)separation, existed [3,61]. The hybrid therapeutic strategy, in which only one injured structure was treated surgically and one structure conservatively, ensured a more stable overall situation for the CCI [62,63].

However, functional post-operative treatment is usually reserved for cases, in which both injuries of the CCI have been surgically stabilized, a procedure that we would recommend for an improved follow-up treatment and early joint mobilization [37,59]. Cases in which it is no longer possible to fix a structure in the direction of the thorax and upper extremity result in, from a functional point of view, a floating clavicle [9,10,17,37]. Therefore, this injury entity should then be viewed as a surgical indication [5,17,37]. This process is the only way to avoid complicated healing processes and, for example, malunion, re-dislocation, and/or persistent displacement as in cases in which a mono-injury is wrongly assumed, the CCI is underestimated, and secondary delayed operations occur [5,16]. Whether and which combinations of surgical procedures have a synergistic effect are reserved for future investigations. Analogous to the still ambiguous therapy decisions concerning Rockwood III ACJD even after extensive research, this ambiguity also applies to CCI in which a large number of surgical techniques have shown that no ideal treatment modality exists that has been capable of prevailing so far with obviously superior outcomes [55].

Since the routine data cannot be traced back due to the anonymized data set, no statements can be made concerning the classification of individual injuries according to Rockwood or Allman, Jäger and Breitner, or Bakir and the study cannot be compared to these existing clavicle fracture classifications [20,23,24,25]. Therefore, a certain bias in the synopsis of our classification with big data analysis could exist. In our opinion, CCIs with mild and incomplete CJIs play a subordinate role, as they did not lead to complete instability from a functional point of view. On the other hand, current studies show that, depending on the diagnosis, a misjudgment is also possible in the classification of ACJD or LCF, both as under- or over-grading [64,65]. It remains to be mentioned that due to a considerable inconsistency in physicians’ classifications, clavicle fracture classification systems in general have previously been unreliable and, therefore, of limited value [65]. This fundamental doubt as to whether newer classification systems would fare any better cannot be dispelled without further studies [47]. An evaluation of our new classification with regard to whether there is better reliability and validity is planned. The coding errors, which make distortion in both directions possible in the sense of under- and over-coding, must be considered a possible major restriction [24]. As in all registry analyses, this might be an important limitation since the missing opportunity for double-checking between coding and radiologic diagnostics means that the coding quality is directly associated with the data accuracy [1,24]. Another limiting factor is that retrospective studies never allow conclusions about the causality of associations, which underlines the importance of future research in this field of CCI.

Overall, the new classification in addition to the therapeutic recommendations are the basis and prerequisites for standardizing CCI treatment. Although the CCIs only account for a small proportion of clavicle injuries, they should not be ignored due to the high number of clavicle injuries in general. Bipolar injuries seem to be predestined to be a missing injury, at least partially on one pole, which could then lead to further complications on this overlooked pole. Future studies are therefore necessary, and the classification system should also be validated. This study should be carried out with a sufficiently large sample size to compensate for the low incidence. A crucial question in this context is whether all types of CCIs consist of equally severe injuries and can therefore be treated with the same priority or whether one type is to be regarded as more serious. For this purpose, clinical analyses should be carried out for validation.

## 5. Conclusions

Overall, our investigations show that CCIs, which have hardly been reported and have not been recognized to date, play a more important role in injuries to the shoulder girdle than previously assumed. Bipolar CCIs occur much less frequently than the monopolar form. According to our recommendation, surgical therapy tends to be the first option. To the best of our knowledge, no classification or therapy standard can be found in the literature yet. The operations published so far are heterogeneous as we describe in a review for the first time. Therefore, further investigations should be aimed at developing a uniform treatment regimen. In general, this work could serve as a basis for placing a special focus on excluding further contra-polar injuries in everyday clinical practice in cases in which a clavicle injury has occurred. This step is particularly important in the case of SCJD in the most commonly affected groups in young and middle adulthood.

## Figures and Tables

**Figure 1 jcm-10-05764-f001:**
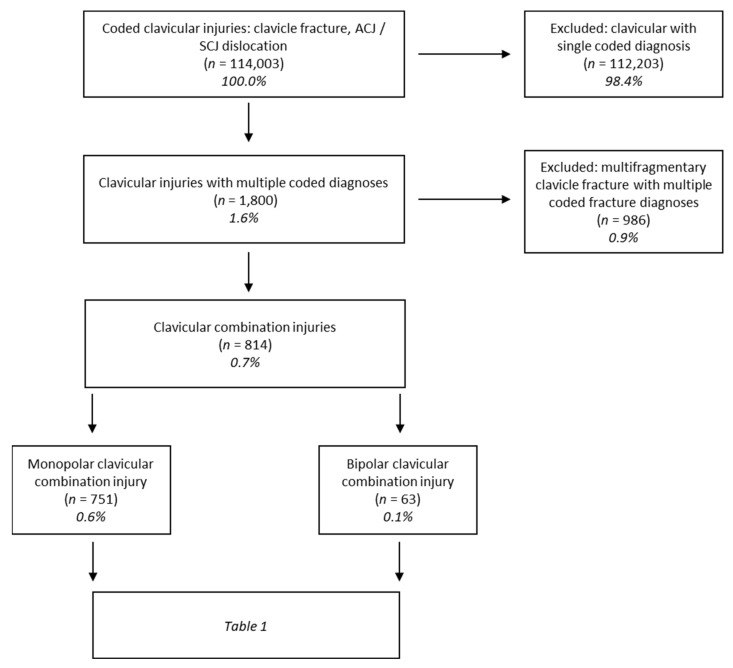
Flowchart of the distribution of clavicular combination injuries (CCI) in 2012–2014 in relation to all clavicle-associated shoulder girdle injuries. CCI represent 0.7% of these injuries with the bipolar type accounting for 0.1%. *n* = number of patients; % = percentage of all shoulder girdle injuries involving the clavicle.

**Figure 2 jcm-10-05764-f002:**
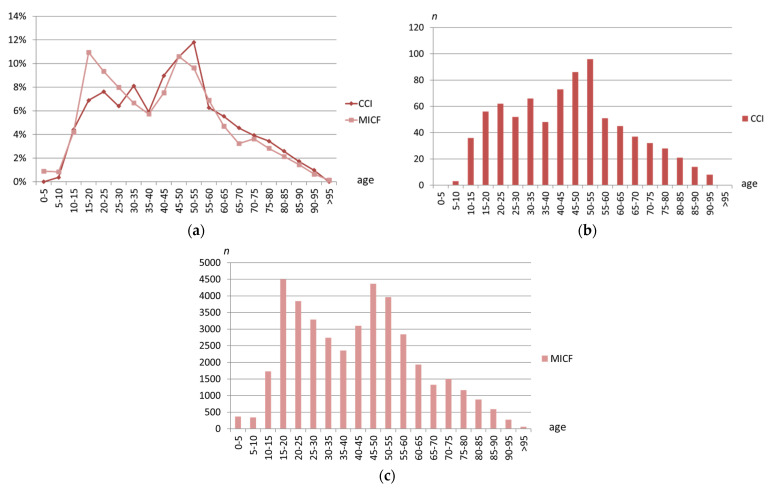
Age distribution of clavicular combination injuries (CCI) in comparison to midshaft clavicle fractures (MICF). Both entities are presented together in a diagram as an overview (**a**) for CCI only (**b**) and for MICF only (**c**) each in detail. The age data are divided into ranges of 5-year periods. The data are shown as a percentage of the respective total injury entity (**a**) and as absolute numbers (**b**,**c**). *n* = number of patients.

**Figure 3 jcm-10-05764-f003:**
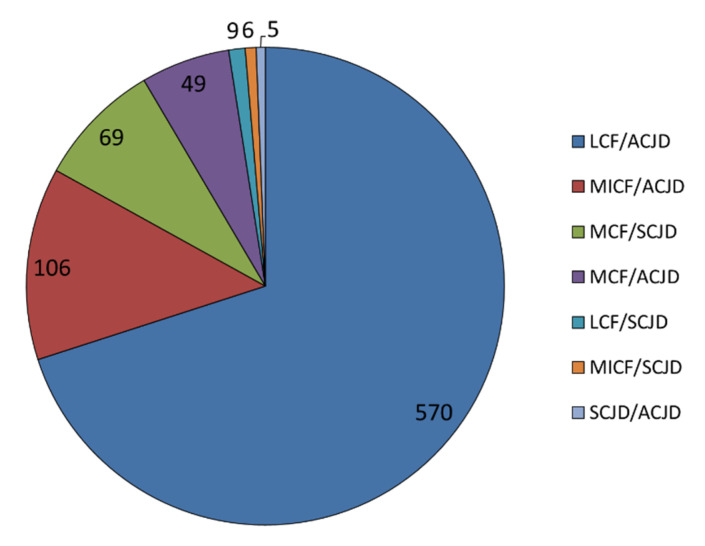
Distribution of all variants of combinations in relation to all clavicular combination injuries (CCI). The respective entities are demonstrated as absolute numbers. SCJD = sternoclavicular joint dislocations; MCF = medial clavicle fracture; MICF = midshaft clavicle fracture; LCF = lateral clavicle fracture; ACJD = acromioclavicular joint dislocation; *n* = number of patients.

**Figure 4 jcm-10-05764-f004:**
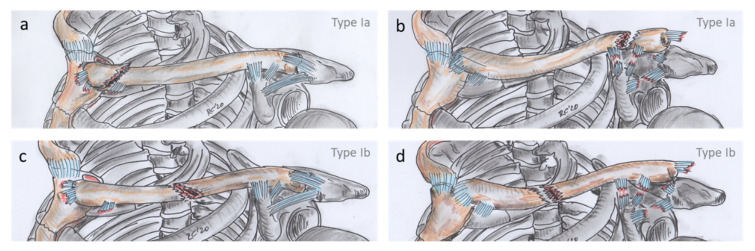
Classification of clavicular combination injuries, showing the more common type I with its possible combinations. All of these type I entities are monopolar injuries. Type Ia are monopolar injuries directly at their respective pole, which can consist of a combination of medial clavicle fracture and sternoclavicular joint dislocation (**a**), and of lateral clavicle fracture and acromioclavicular joint dislocation (**b**). Type Ib are monopolar injuries including a midshaft clavicle fracture (MICF), which can consist of a combination of sternoclavicular joint dislocation and MICF (**c**), and of acromioclavicular joint dislocation and MICF (**d**).

**Figure 5 jcm-10-05764-f005:**
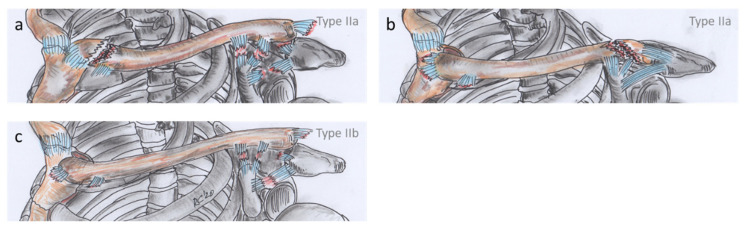
Classification of clavicular combination injuries, showing the less common type II with its possible combinations. All of these type II entities are bipolar injuries. Type IIa are ligamento-osseous bipolar injuries, which can consist of a combination of medial clavicle fracture and acromioclavicular joint dislocation (**a**), and of lateral clavicle fracture and sternoclavicular joint dislocation (**b**). Type IIb are purely ligamentous bipolar injuries, which means a combination of sternoclavicular and acromioclavicular joint dislocation (**c**).

**Figure 6 jcm-10-05764-f006:**
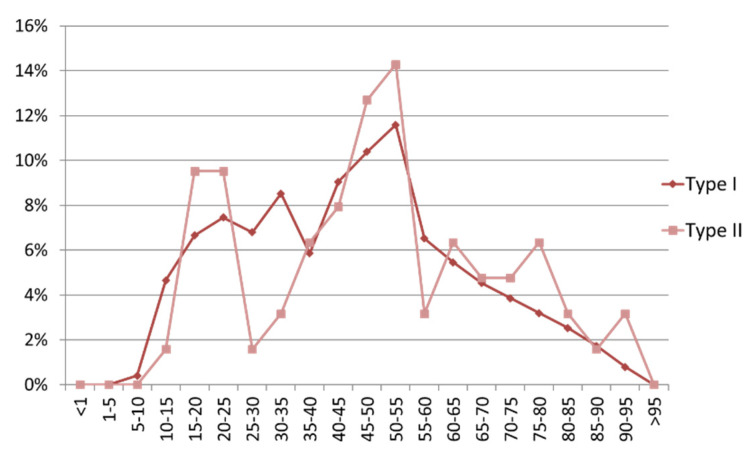
Age distribution of clavicular combination injuries (CCI), comparison of type I and type II injuries. The age data are divided into ranges of 5-year periods. The data are shown as a percentage of the respective total type. The absolute numbers for type I injuries were *n* = 751 and for type II injuries *n* = 63 patients.

**Figure 7 jcm-10-05764-f007:**
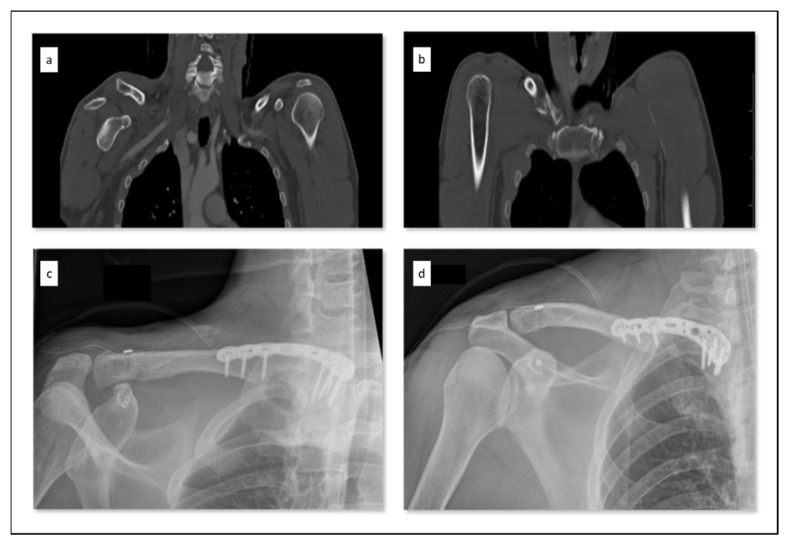
Example of recommended treatment strategy for a clavicular combination injury (CCI) type IIa in the case of bipolar injury to the shoulder girdle. A male patient with 45 years of age was injured because of a bicycle accident. The computed tomography showed a combination of an acromioclavicular joint dislocation (**a**) and an ipsilateral medial clavicle fracture (**b**). This CCI was surgically treated with open reduction and internal plate fixation of the medial clavicle (Variax, Stryker) and acromioclavicular joint reconstruction (Tightrope, Arthrex) in one session (**c**,**d**).

**Table 1 jcm-10-05764-t001:** Cross table of all variants of clavicular combination injuries in relation to each other.

		Secondary Diagnosis				
		SCJD	MCF	MICF	LCF	ACJD
Main diagnosis	SCJD	-	16	1	1	2
	MCF	53	-	268	66	36
	MICF	5	275	-	151	84
	LCF	8	63	163	-	450
	ACJD	3	13	22	120	-

The single diagnoses are demonstrated as main and secondary diagnosis and as the number of injured patients to the respective combination. The outer circle represents the clavicular combination injuries, which are in bold. The inner circle represents the multi-fragmentary clavicle fractures, which are in smaller letters and in italics. SCJD = sternoclavicular joint dislocations; MCF = medial clavicle fracture; MICF = midshaft clavicle fracture; LCF = lateral clavicle fracture; ACJD = acromioclavicular joint dislocation.

**Table 2 jcm-10-05764-t002:** The proportions of clavicular combination injuries (CCI) in relation to the respective injury entity differed significantly from one another (*p* < 0.001 to *p* = 0.023).

	SCJD	MCF	MICF	LCF	ACJD
*n* total	676	13,588	41,156	28,887	29,696
*n* CCI	89	118	112	579	730
CCI/total	13.2%	0.9%	0.3%	2.0%	2.5%
*p* value	<0.001	0.023	<0.001	<0.001	<0.001

*n* = number of patients; SCJD = sternoclavicular joint dislocations; MCF = medial clavicle fracture; MICF = midshaft clavicle fracture; LCF = lateral clavicle fracture; ACJD = acromioclavicular joint dislocation.

**Table 3 jcm-10-05764-t003:** Classification of clavicular combination injuries (CCI), showing the distribution of all types and entities of combinations in relation to all CCI. Proposed classification of clavicular combination injuries.

Table	Polarity	Proportion of CCI	Subtype	Definition Subtype	Proportion of CCI	Entity of Combination		Proportion of CCI
Type I	monopolar	92.3%	Type Ia	monopolar direct	78.5%	L	LCF/ACJD	70.0%
						M	MCF/SCJD	8.5%
			Type Ib	monopolar and midshaft	13.8%	L	MICF/ACJD	13.0%
						M	MICF/SCJD	0.7%
Type II	bipolar	7.7%	Type IIa	ligamento-osseous	7.1%	MCF/ACJD		6.0%
						LCF/SCJD		1.1%
			Type IIb	ligamentous	0.6%	SCJD/ACJD		0.6%

SCJD = sternoclavicular joint dislocations; MCF = medial clavicle fracture; MICF = midshaft clavicle fracture; LCF = lateral clavicle fracture; ACJD = acromioclavicular joint dislocation; L = lateral; M = medial.

**Table 4 jcm-10-05764-t004:** Therapeutic algorithm of clavicular combination injuries (CCI), showing the therapy recommendations/suggestions (and alternative therapy options) based on the classification and related to the corresponding subtype.

Type	Polarity	Subtype	Definition Subtype	Entity of Combination		Therapy Recommendations
Type I	monopolar	Type Ia	monopolar direct	L	LCF/ACJD	- Hook plate - Lateral clavicle locking plate + TightRope ACJ
				M	MCF/SCJD	- Contralateral lateral clavicle locking plate at medial + TightRope SCJ - (Temporary arthrodesis plate SCJ)
		Type Ib	monopolar and midshaft	L	MICF/ACJD	- Long hook plate - Clavicle locking plate + TightRope ACJ - (hook plate + clavicle locking plate)
				M	MICF/SCJD	- Clavicle locking plate + TightRope SCJ - (Arthrodesis SCJ long plate)
Type II	bipolar	Type IIa	ligamento-osseous	MCF/ACJD		- Clavicle locking plate + TightRope ACJ (Figure 7) - (Clavicle locking plate + hook plate) - (Contralateral lateral clavicle locking plate at medial + hook plate) - (Contralateral lateral clavicle locking plate at medial + TightRope ACJ)
				LCF/SCJD		- Lateral clavicle locking plate + TightRope SCJ - Hook plate + TightRope SCJ - (Lateral clavicle locking plate + temporary arthrodesis plate SCJ) - (Hook plate + temporary arthrodesis plate SCJ)
		Type IIb	ligamentous	SCJD/ACJD		- TightRope SCJ + hook plate - Temporary arthrodesis plate SCJ + TightRope ACJ - (TightRope SCJ + TightRope ACJ)

SCJD = sternoclavicular joint dislocations; MCF = medial clavicle fracture; MICF = midshaft clavicle fracture; LCF = lateral clavicle fracture; ACJD = acromioclavicular joint dislocation; L = lateral; M = medial.

## Data Availability

Restrictions apply to the availability of these routine data. Data was obtained from the German Federal Statistical Office, was used under license for the current study, and are available from the authors with the permission of the German Federal Statistical Office.

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
