# Peer review of "Monopolar and Bipolar Combination Injuries of the Clavicle: Retrospective Incidence Analysis and Proposal of a New Classification System"

_jcm, 2021, doi:10.3390/jcm10245764_

Round 1

Reviewer 1 Report

The title is complicated. A more comprehensive title is necessary such as: Monopolar and bipolar fractures of the clavicle: Retrospective incidence analysis and proposal of a new classification system.

Line 17. Define combined injuries

Line 18. Delete “as floating clavicles”

Line 20. Delete routine data according to,

Line 20. What is routine data?

Line 24. Unclear statement” with highest proportion in SCJD (13.2% of all SCJD)”

Lines 25 and 26. The meaning of this sentence is evading the reader. Be more specific.

Line 32 Keywords: clavicular combination injury. This term is not included in the MESH term index

The abstract is extended and not focused. Please re-write the abstract.

Paragraphs 1 and 2 should be merged.

Line 36. Unclear “and the study situation is analogous to the frequency of the various entities”

Line 39-40 merge with the following sentence

Line 45. Delete -like

Lines 46-47. “It is usually assumed”. This statement is based only on a case report. Restructure and provide more valid references

Line 48. mono-injury, is not a valid term

Lines 56-60. This paragraph is not essential.

Lines 66-68. Merge with the previous paragraph

Materials and methods. Although this is a database study it is not clear how the combined injuries were diagnosed. The classification was based on radiographs, CT or MRI scans?

Lines 82-92. These lines belong to paragraph 3.2

Line 112. Pure, is not a valid term

Figure 1. Prism of distribution, is not a valid expression

Lines 163-172. Need to be more accurate and focused. Difficult to follow.

A figure with radiographic and CT scan images documenting the presence of the combined injuries is necessary. In this classification all theoretically possible combinations are presented, but this does not mean that they can occur.

Figure 4a. the SCJ dislocation is not shown

Discussion. It is unacceptably long and tiring. Should be reduced to 2 pages the most.

A paragraph more clearly stating the limitations of the study, which is based on ICD diagnoses should be included in the study.

Reviewer 2 Report

Overall

Summary: In this study, the authors retrospectively analyzed the database and proposed a new classification system for combined clavicular injury. T

Abstract

  1. Introduction

            Line 39 Please spell the full name of CCI when first mentioned.

  1. Materials/methods

Line 71 you use the data from 2012 to 2014. Why not use the more updated data.

  1. Results

Line 145

What did medial clavicular injury (MCI) mean? How do you get the case number for MCI. Did MCI include medial clavicle fracture an sternoclavicular dislocation? You should mention it in the section of method. Same issue existed for lateral clavicular injury (LCI).

Line 147

“In particular, the proportion of explicitly contra-polar lateral injuries in MCI with 0.4% (63/14,264) compared to the proportion of medial contra-polar injuries in LCI with 0.1% (63/58.583) showed a significant difference (p < 0.001). “

What did contra-polar lateral injury / medial contra-polar injury mean? You should define it in the section of method. Where did the actual number come from? Table 1? Table 2? Please revise this section to make it clear to read.

Line 152

“The proportions of CCI in relation to the respective injury entity differed significantly from one another (p < 0.001–0.023) with SCJD showing by far the highest proportion of combination injuries with more than every 10th case of SCJ injury (Table 2)”

   This statement was hard to understand. Please give each p value to corresponding comparison in the table.

  1. Discussion
  • What’s the clinical relevance? Did this new classification provide any new information or guidance for treatment? I don’t think it really helpful for treatment guidance unless it was correlated treatment outcome
  • The coding error is a main issue for retrospective database study, especially for your study since you tried to propose a new classification system based on the potential biased information. It may be more reasonable to do validation study first using the single center database, and then re-exam the results in the nationwide database,
  1. Conclusion

No comments

Round 2

Reviewer 1 Report

The revised version fulfils most of the comments made in the previous paper review.